# Knowledge, attitude, practices, and determinants of them toward tuberculosis among social media users in Bangladesh: A cross-sectional study

**Sultan Mahmud**[1], **Md Mohsin**[2]*, **Saddam Hossain Irfan**[2], **Abdul Muyeed**[3], **Ariful Islam**[4]

1 International Centre for Diarrhoeal Disease Research, Bangladesh (ICDDR'B), Dhaka, Bangladesh, 2 Department of Applied Statistics, Institute of Statistical Research and Training, University of Dhaka, Dhaka, Bangladesh, 3 Department of Statistics, Jatiya Kabi Kazi Nazrul Islam University, Trishal, Mymensingh, Bangladesh, 4 Department of Statistics, University of Dhaka, Dhaka, Bangladesh

* mmohsin@isrt.ac.bd

## Abstract

**Data Availability Statement:** The data that support the findings of this study are openly available at

### Objectives

Tuberculosis (TB) is an infectious disease that causes thousands of deaths in Bangladesh. Bangladesh is one of the 30 high TB burden countries. In this study, we aimed to assess the knowledge, practices, and attitude toward TB, and to determine the factors associated with them among people who have internet access in Bangladesh. Design, Setting, and Participant: A web-based anonymous cross-sectional survey was conducted from May 20 to August 10, 2021, among people (age> = 18 years) who have internet access in Bangladesh. A comprehensive consent statement was included at the beginning of the survey and informed consent was taken.

### Outcome measures

This study's outcomes of interest were respondents' adequate knowledge, good practices, and positive attitudes toward TB and were coded binarily. The association between respondents' socio-demographic factors and knowledge, attitude, and practices toward TB was inspected using the Chi-square test and Multivariable logistic regression model.

### Results

Among 1,180 respondents, 58.64% were males, and 62.37% were married. The majority of the participants (78.28%) were aged between 18 to 44 years. Overall adequate knowledge, favorable attitudes, and good practices about TB were found respectively in 47.8%, 44.75%, and 31.19% of the people with internet access in Bangladesh. Almost the same sets of associated factors were found to influence adequate knowledge, favorable attitudes, and good practices toward TB among social media users in Bangladesh. Males, young,

Open Science Framework doi:10.17605/OSF.IO/
A9MR2 (https://osf.io/a9mr2/).

**Funding:** The author(s) received no specific
funding for this work.

**Competing interests:** The authors have declared
that no competing interests exist.

unmarried, social media users with higher education, and urban social media users were
more likely to have adequate knowledge, favorable attitudes, and good practices toward TB.

## Conclusion

Policymakers need to design programs and interventions to improve knowledge, attitudes,
and practices toward TB in Bangladesh with a particular focus on females, young and older
people, people who live in rural areas, and illiterate/less educated people. Social media can
be a powerful medium for disseminating scientific facts on TB and other diseases.

## Introduction

Tuberculosis (TB) is an infectious bacterial disease caused by Mycobacterium tuberculosis
(MTB) [1]. TB remains a severe health problem worldwide, despite a tremendous performance
in controlling the disease, with an estimated 37 million lives saved by improved diagnosis and
treatment since 2000 [2]. Per a 2015 global health report, tuberculosis (TB) is the leading cause
of morbidity and mortality globally, ranking alongside the human immunodeficiency virus
(HIV) [3]. Globally, 10.4 million people were reported to have contracted tuberculosis in 2015,
with 1.8 million people dying from the disease [4]. However, developing countries bear the
brunt of tuberculosis's impact. In 2015, almost 95% of the estimated 1.8 million TB deaths
occurred in low- and middle-income countries [3].

Among those who were infected with tuberculosis in 2019, 79% were from 30 high-burden
countries [5]. In 2019, Bangladesh was one of the 30 countries with the highest TB burden,
accounting for 3.6% of the global total [6]. According to the Global TB Report 2020, 0.7% of
new cases and 11% of previously treated patients in Bangladesh were positive for multidrug-
resistant tuberculosis (MDR-TB), which has an incidence rate of 2.0 per 100,000 people [5].

Even though tuberculosis is a preventable and treatable disease, the situation in Bangladesh
has remained essentially constant over the years, with moderate progress and no signs of a
breakthrough in the near future [6]. Bangladesh established End TB goals, including a 95%
reduction in TB mortality and a 90% reduction in TB incidence by 2035 compared to 2015 lev-
els, with intermediary goals set for 2020, 2025, and 2030 [7]. The Stop TB Partnership has
issued TB diagnosis and treatment targets for Bangladesh for 2018–2022 as a result of the
United Nations High-Level TB Meeting (UNHLM). To meet the UNHLM's cumulative five-
year TB targets, Bangladesh must raise diagnosis and treatment by 45% above the total notifi-
cations reported during the pre-UNHLM five-year period (2014–2017) [8].

The National Tuberculosis Control Program (NTP) has chosen the Directly Observed
Treatment, Short-course (DOTS) technique to lessen this burden, which is predominantly
given through government-run health institutions [9]. However, considerable impediments to
implementation exist, mainly due to insufficient infrastructure and suitable health workers
[10]. Therefore, the World Health Organization recommends that national TB Control Pro-
grams use an Advocacy, Communication, and Social Mobilization (ACSM) framework to
address these issues. This strategy framework targets four significant issues: enhancing case
detection and treatment adherence, eliminating stigma and prejudice, empowering tuberculo-
sis patients, and mobilizing the resources and political commitment needed to combat the dis-
ease [11].

Despite the efforts, the expected degree of improvements in controlling the TB crisis has
not been made yet in Bangladesh [12]. One of the primary challenges in preventing,

controlling, and eliminating tuberculosis is a lack of awareness and knowledge and a negative attitude about the disease [1]. There are also a lot of misconceptions concerning the etiology and mode of transmission of TB in Bangladesh [13, 14]. A lack of understanding about tuberculosis and old misconceptions are linked to delays in case detection and treatment for TB [15, 16]. The widespread prejudice toward TB/HIV patients, misapprehension of transmission of TB and other infectious diseases, and poor knowledge about the treatment of infectious diseases are serious restrictions in achieving millennium development goals related to TB and other infectious diseases [17–19].

The authors of this study understand that the lack of knowledge, misconceptions, and bad practices among the general population, around 43% of them were active internet users in 2021 [20], could be the reasons for slow and unsatisfactory progress in the fight against TB in Bangladesh. Unfortunately, there is a lack of large-scale studies in Bangladesh that explore the knowledge and attitude about TB and practices to prevent it. Therefore, this study aimed to investigate knowledge, attitudes, and practices toward TB among the general population who have access to the internet. Also, this study explores the risk factors associated with poor knowledge, attitude, and practices toward TB among the participants. The government should renew its commitment to national tuberculosis control activities based on data-driven, effective methods to meet the stipulated goals. The findings of this study would be a significant help for the government and policymakers in this regard.

## Methods

### Study design and study participants

This study was a cross-sectional approach to collect data regarding knowledge, attitude, and practices about tuberculosis in Bangladesh. It was an online anonymous, self-interviewed survey conducted from May 20 to August 10, 2021. People aged 18 and over and living in Bangladesh were eligible to participate in this survey. In the beginning, there was a section describing the study's objective, the idea of the questionnaire, assurance about the respondents' confidentiality, and the study's voluntary nature. It was also mentioned that participants could skip a question if it seemed sensitive. The online survey started with the respondents' informed consent and the eligibility check. The voluntary participants were also requested to share the survey link with their connections after completion. An online survey link (KoBoToolbox) was shared with almost 4000 internet users in Bangladesh through social media (FB, WhatsApp, Instagram, Email, etc.). A total of 1,205 (response rate was 30%) people filled out and submitted their responses; among them, 25 of the respondents were not eligible (either aged less than 18 or living outside of Bangladesh) for this study.

### Sample size

In this study, we aimed to examine the knowledge, practices, and attitude toward TB and their associated factors among the general population in Bangladesh. We did not find previous literature from Bangladesh that examined the knowledge, practices, and attitude toward TB and their associated factors among the general population. For calculating the desirable sample size, we assume that 50% of the general population has adequate knowledge about TB, good practices, and a favorable attitude toward TB. Using an online sample size calculator [21], we found that this study requires a sample size of 591 to represent a population size of 164,689,383 [22] with 5% absolute precision, 95% confidence, and an expected response rate of 65%.

## Instruments

The study instrument/questionnaire was adapted from previously developed validated questionnaires and translated into Bangla [1, 23, 24]. Then, the final questionnaire (S1 Questionnaire) was validated by several experts and pilot surveys. The structured questionnaire was made of 3 main sections: (i) Background characteristics of the respondents; (ii) Risk behaviors related to tuberculosis; (iii) Knowledge of tuberculosis; (iv) Attitudes toward tuberculosis; and (v) Practices of tuberculosis.

At the outset of the survey, we checked the aptness of the participants by asking two questions, "How old are you (in years)?" and "Do you currently live in Bangladesh?" We also added respondents' socio-demographics and some personal details in this section (not identifiable). The socio-demographic details were gender, current marital status, religion, educational qualification, and socioeconomic details were monthly household income level, occupational status, residence, etc. The second part of the survey questionnaire contained questions linked to risk behaviors related to tuberculosis, including diabetes status, smoking status, the status of drinking alcohol in the last three months, the status of exposure to indoor cooking smoke, etc. Demographic covariates of this study were categorized in the following way: Residence: Rural, Urban; and Religion: Muslims, Hindu, Buddhists/Cristian; Sex: Female, Male; Age (year): 18–29, 30–44, 45–59, 60–74, 75+; Marital Status: Married, Unmarried, Others (Divorced, Widowed, Separated); Education: Less or equal SSC (10th grade), HSC (12th grade), undergraduate, Master's or higher, Never been to school; Last month income (Taka): Less than 10 thousand, 11–20 thousand, 21–30 thousand, 31–40 thousand, greater than 40 thousand, No income; Occupation: Business, Housewife, Govt. employee, Non-govt. employee, Unemployed, Self-employed, Student. The level of knowledge about tuberculosis was assessed by asking a series of questions under a few sub-segments, "Source of knowledge on TB", "Knowledge about TB causes", "Knowledge about the transmission of TB", "Knowledge about symptoms of TB", and "Knowledge about availability of TB treatment". In addition, the participants were also asked a series of questions to assess the level of attitude and practices regarding tuberculosis.

## Consent and ethical considerations

At the outset of the survey, a section described the study's eligibility, aims, the questionnaire's concept, assurances regarding respondents' confidentiality, and the study's voluntary nature. Additionally, it was indicated that participants could omit a question if it appeared to be sensitive. This study was reviewed and waived the requirement of an IRB approval by the Ethical Review Committee, Faculty of Biological Science and Technology, University of Science and Technology, Jashore, Bangladesh. Because this was an anonymous online survey, it was voluntary, and it did not include any clinical operations.

## Data management

A standard procedure was adopted to minimize the data collection errors and to ensure the high quality of information. A Stata program was developed for monitoring the time-to-time data collection progress. The inconsistency and duplicate checking were also part of the program. Moreover, the questionnaire was programmed in KoBo Toolbox in a way that automatically generates a device id for each of the devices from which participants completed the survey. The duplicate submissions were identified and dropped by using the device id which is expected to be unique. A complete and clean data set was used for the final analysis.

## Statistical analysis

This study's primary outcome of interest was respondents' adequate knowledge, good practices, and positive attitudes towards TB. Participants' knowledge of the cause, mode of transmission, signs, and symptoms, and treatment availability of TB was coded as "1" and labeled as "adequate knowledge" if the respondent correctly answered $\geq$ 50% ($\geq$ 7 questions out of the total 14) of questions. Otherwise, participants' knowledge was coded as "0" and labeled as "poor knowledge". The overall participants' attitude towards TB was defined as "Favorable attitude" and coded as "1" if the respondent correctly answered $\geq$ 3 questions out of the total 5 and otherwise defined as "Unfavorable attitude" and coded as "0". We used two questions to assess respondents' practice toward TB (Q1: If you had symptoms of TB, at what point would you go to the health facility? and Q2: If you had symptoms of TB, where will you go for TB treatment?). The overall respondents' practices toward TB were defined as "Good practice" if the respondent correctly answered both questions otherwise define as "Bad practice".

The exploratory analysis (frequencies analysis, means, median, bivariate analysis) was done to check socio-demographic characteristics. The statistical significance of the correlation between socio-demographic factors and knowledge of the respondents and their practice and attitude towards TB was inspected using the Chi-square test. All the significant factors at a 10% level of significance in the Chi-square test were included in the univariate logistic regressions [25]. We did so to recheck the association between socio-demographic factors and knowledge of the respondents and their practice and attitude toward TB. The adjusted odds ratios (AOR) were also calculated using multivariable logistic regression [26, 27] with a 95% confidence interval (CI). All the analyses were done by using the Statistical package STATA version 16.0.

## Results

### Socio-demographic characteristics

More than four thousand people were invited to participate in the survey through online platforms (WhatsApp, Messenger, Email, Linkedin, etc.). A total of 1,180 (30% response rate) people submitted the self-consent completed surveys. The socio-demographic characteristics of the respondents are presented in Table 1. The majority of the respondent tended to be male (58.64%), aged between 18 to 44 years (78.28%), and married (62.37%). About one-third of the respondents (34.80%) had a Master's or higher degree. Most of the respondents were Muslim (87.12%), and living in rural areas (56.61%). Almost half of the respondents were students (44.75%).

### Knowledge about tuberculosis and associated factors

All the respondents confirmed that they heard about TB. In response to a multiple-response question, we found that the source of information about TB for 85.76% of the respondents was TV/Radio/Newspaper (Fig 1). The second major source was leaflets/Poster/Signboard/ Billboard (66.78%). Nearly half of the respondents received information from health professionals, and 44.4% received it from the internet. Religious leaders/teachers were the sources of information for 43.05% of the respondents. A similar proportion of the participants (41.36%) learned about TB from friends/relatives/family members. Exposure to TB treatment and inmates suffering from TB were the sources of TB for 27.80% and 17.29% of the participants. Table 2 shows that a large proportion (71%) of the participants knew that TB germ or Bacteria is the major cause of TB. Nearly 42% of the respondents knew the correct transmission mode of TB (TB can be spread from person to person through the air when coughing or sneezing).

**Table 1. Distribution of socio-demographic characteristics of respondents.**

| Variable | Labels | N (%) |
| --- | --- | --- |
| Gender | Male | 692 (58.64) |
| | Female | 488 (41.36) |
| Age | 18–29 | 404 (34.24) |
| | 30–44 | 520 (44.07) |
| | 45–59 | 240 (20.34) |
| | 75+ | 16 (1.36) |
| Marital status | Married | 736 (62.37) |
| | Unmarried | 328 (27.80) |
| | Divorced/Widowed/Separated | 116 (9.83) |
| Education | Less or equal HSC ($\leq 12^{th}$ grade) | 332 (28.14) |
| | Undergraduate | 408 (34.58) |
| | Master's or higher (Graduate) | 440 (37.29) |
| Income | Less than 30,000 | 276 (23.39) |
| | 30,000–45,000 | 176 (14.92) |
| | 46,000–60,000 | 160 (13.56) |
| | 61,000–75,000 | 280 (23.73) |
| | 76,000 and above | 288 (24.41) |
| Occupation | Service holder (govt/private) | 324 (27.46) |
| | Entrepreneur/business | 172 (14.58) |
| | Student | 528 (44.75) |
| | Housewife/Retired/Unemployed/Other | 156 (13.22) |
| Religion | Islam | 1,028 (87.12) |
| | Hinduism | 134 (11.36) |
| | Buddhists/Cristian | 18 (1.53) |
| Region | Urban | 512 (43.39) |
| | Rural | 668 (56.61) |

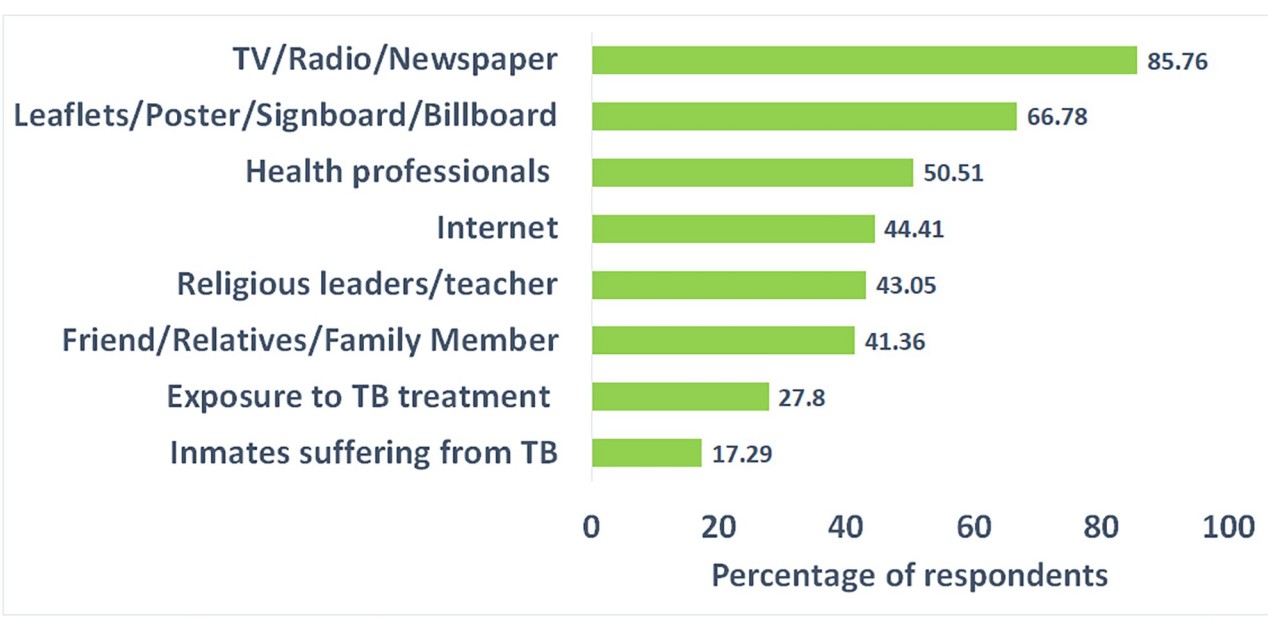

**Fig 1. Source of information of the respondents about TB.**

**Table 2. Knowledge of respondents on TB cause, transmission, signs & symptoms, treatment, and attitudes and practices toward TB among participants.**

| Question | Label | N (%) |
|---|---|---|
| **Knowledge about TB causes** | | |
| What is the primary cause of TB? | TB germ /Bacteria | 828 (70.41) |
| | Virus | 188 (15.99) |
| | Cold wind | 8 (0.68) |
| | Smoking | 36 (3.06) |
| | Spoiled soil (soil with a bad odor) | 0 (0) |
| | Poor hygiene Alcohol | 4 (0.34) |
| | Inherited | 0 (0) |
| | Don't know | 112 (9.52) |
| **Knowledge about the transmission of TB** | | |
| TB is spread from person to person through the air when coughing or sneezing? | Yes | 488 (41.36) |
| | No | 672 (56.95) |
| | Don't know | 20 (1.69) |
| Can TB be transmitted by sharing utensils? | Yes | 584 (49.49) |
| | No | 556 (47.12) |
| | Don't know | 40 (3.39) |
| Can TB be transmitted through food? | Yes | 572 (48.47) |
| | No | 600 (50.85) |
| | Don't know | 8 (0.68) |
| Can TB be transmitted through sexual contact? | Yes | 624 (52.88) |
| | No | 532 (45.08) |
| | Don't know | 24 (2.03) |
| What is the most common site for TB infection in the body? (Only one answer) | Lungs | 952 (80.68) |
| | Glands | 140 (11.86) |
| | Brain | 0 (0) |
| | Bones | 4 (0.34) |
| | Others (specify) — | 0 (0) |
| | Don't know | 84 (7.12) |
| **Knowledge about symptoms of TB** | | |
| A person who is infected with TB coughs for several (more than 3) weeks? | Yes | 488 (41.36) |
| | No | 684 (57.97) |
| | Don't know | 8 (0.68) |

(*Continued*)

**Table 2.** (Continued)

| Question | Label | N (%) |
|---|---|---|
| A person who is infected with TB has a persistent fever | Yes | 404 (36.33) |
| | No | 700 (62.95) |
| | Don't know | 8 (0.72) |
| A person who is infected with TB sweats during the night | Yes | 512 (43.39) |
| | No | 652 (55.25) |
| | Don't know | 16 (1.36) |
| A person who is infected with TB has pain in the chest or back | Yes | 512 (43.39) |
| | No | 664 (56.27) |
| | Don't know | 4 (0.34) |
| Weight loss is one of the symptoms of TB | Yes | 492 (41.69) |
| | No | 668 (56.61) |
| | Don't know | 20 (1.69) |
| **Knowledge about the availability of TB treatment** | | |
| Is TB management available free of cost in Bangladesh? | Yes | 872 (73.9) |
| | No | 176 (14.92) |
| | Don't know | 132 (11.19) |
| Is TB curable? | Yes | 1020 (86.44) |
| | No | 140 (11.86) |
| | Don't know | 20 (1.69) |
| **Attitude towards TB** | | |
| In your opinion, how serious disease is TB? | Very serious | 936 (79.32) |
| | Somewhat serious | 112 (9.49) |
| | Not very serious | 132 (11.19) |
| Do you afraid to get infected with TB? (chose only one) | Yes | 544 (46.1) |
| | No | 624 (52.88) |
| | Don't know | 12 (1.02) |
| Will you keep it secret when any family member gets TB? | Yes | 976 (82.71) |
| | No | 196 (16.61) |
| | Don't know | 8 (0.68) |
| Would you be willing to work with someone previously treated for TB? | Yes | 528 (44.75) |
| | No | 648 (54.92) |
| | Don't know | 4 (0.34) |

(*Continued*)

**Table 2.** (Continued)

| Question | Label | N (%) |
|---|---|---|
| What would be your reaction if you found out that you have TB? (chose only one) | Go to pharmacy | 536 (45.42) |
| | Go to a health facility | 448 (37.97) |
| | Got to a traditional healer | 144 (12.2) |
| | Pursue other self-treatment options (herbs, etc.) | 44 (3.73) |
| | Others (specify) | 8 (0.68) |
| **Practice toward TB** | | |
| If you had symptoms of TB, at what point would you go to the health facility? (choose only one) | When treatment on my own does not work. | 208 (17.63) |
| | When symptoms that look like TB signs last for 3–4 weeks. | 336 (28.47) |
| | As soon as I realize that my symptoms might be related to TB. | 568 (48.14) |
| | I would not go to the doctor. | 68 (5.76) |
| If you had symptoms of TB, where would you go for TB treatment? (chose only one) | Modern drugs | 832 (70.51) |
| | Herbal Remedies | 148 (12.54) |
| | Home Remedies | 124 (10.51) |
| | Praying /holy water | 44 (3.73) |
| | Don't Know | 32 (2.71) |

Nevertheless, the misconception was observed among a considerable proportion (57%) of the respondents. Almost half of the respondents knew that TB could be transmitted from person to person by sharing utensils and food or by sexual contact. A more significant proportion (81%) of the respondents correctly knew that the lung is the most common site for TB infection in the body. However, the respondent's knowledge about the symptoms of TB was deficient. Out of 1,180 persons, 652 (55.25%) did not know that coughing for several (more than 3) weeks is a common symptom of TB infection. Less than 50% (36.33%, 43.39%, and 41.69%, respectively) of the respondents knew that persistent fever, sweats during the night, and weight loss are TB symptoms. Knowledge about the availability of TB treatment was considerably high among the respondents. Almost 85% of the respondents knew that TB is curable, and 74% knew that TB treatment is available and accessible in Bangladesh.

We observed adequate overall knowledge regarding TB in only 47.8% of the respondents (Table 3). About 68.21% of males and only 18.85% of females had adequate knowledge about TB. The findings from multivariable regression also show that females had a 90% lower chance of having adequate knowledge about TB than their male counterparts. The age of the respondents was also a significantly associated factor for having adequate knowledge regarding TB. Middle-aged people were more likely to have more knowledge about TB. More explicitly, respondents aged 30–40 had a 2.13 times higher likelihood of having adequate knowledge than respondents aged 18–29. However, older respondents (aged 45–59) had around 84% lower chance of having adequate knowledge about TB than the young respondents (aged 18–29). Marital status, education level, and income were strongly correlated with the overall level of knowledge. Around 42% of married and 75.61% of unmarried respondents had adequate knowledge about TB. The odds of having adequate knowledge among the unmarried

**Table 3. Associated factors with knowledge of TB among participants.**

| Factors | Bivariate analysis | | | | | Multivariable analysis | |
|---|---|---|---|---|---|---|---|
| | Adequate knowledge n (%) | Poor knowledge n (%) | P-value | UOR (95% CI) | P-value | AOR (95% CI) | P-value |
| Total | 564 (47.8) | 616 (52.20) | | | | | |
| Gender | | | | | | | |
| Male | 472 (68.21) | 220 (31.79) | < .001 | Ref | | Ref | |
| Female | 92 (18.85) | 396 (81.15) | | 0.11 (0.08–0.14) | < .001 | 0.1 (0.07–0.15) | < .001 |
| Age | | | | | | | |
| 18–29 | 184 (45.54) | 220 (54.46) | | Ref | | Ref | |
| 30–44 | 332 (63.85) | 188 (36.15) | < .001 | 2.11 (1.62–2.75) | < .001 | 2.13 (1.37–3.32) | < .01 |
| 45–59 | 44 (18.33) | 196 (81.67) | | 0.27 (0.18–0.39) | < .001 | 0.16 (0.08–0.31) | < .001 |
| 75+ | 4 (25.00) | 12 (75.00) | | 0.4 (0.13–1.26) | NS | 1.94 (0.19–19.45) | NS |
| Marital Status | | | | | | | |
| Married | 308 (41.85) | 428 (58.15) | | Ref | | Ref | |
| Unmarried | 248 (75.61) | 80 (24.39) | < .001 | 4.31 (3.22–5.77) | < .001 | 4.01 (2.51–6.42) | < .001 |
| Other£ | 8 (6.90) | 108 (93.10) | | 0.1 (0.05–0.21) | < .001 | 0.02 (0–0.06) | < .001 |
| Education | | | | | | | |
| Less or equal HSC (< = 12th grade) | 52 (15.66) | 280 (84.34) | | Ref | | Ref | |
| Undergraduate | 256 (62.75) | 152 (37.25) | < .001 | 9.07 (6.34–12.97) | < .001 | 5.06 (2.93–8.74) | < .001 |
| Master's or higher (Graduate) | 256 (58.18) | 184 (41.82) | | 7.49 (5.27–10.65) | < .001 | 3.71 (2.2–6.27) | < .001 |
| Income | | | | | | | |
| Less than 30,000 | 108 (39.13) | 168 (60.87) | | Ref | | Ref | |
| 30,000–45,000 | 36 (20.45) | 140 (79.55) | | 0.4 (0.26–0.62) | < .001 | 0.27 (0.14–0.53) | < .001 |
| 46,000–60,000 | 48 (30.00) | 112 (70.00) | < .001 | 0.67 (0.44–1.01) | NS | 0.35 (0.18–0.69) | < .01 |
| 61,000–75,000 | 132 (47.14) | 148 (52.86) | | 1.39 (0.99–1.94) | NS | 0.94 (0.54–1.65) | NS |
| 76,000 and above | 240 (83.33) | 48 (16.67) | | 7.78 (5.25–11.52) | < .001 | 9.12 (4.35–19.11) | < .001 |
| Occupation | | | | | | | |
| Service holder (govt/private) | 220 (67.90) | 104 (32.10) | | Ref | | Ref | |
| Entrepreneur/business | 108 (62.79) | 64 (37.21) | < .001 | 0.8 (0.54–1.18) | NS | 0.64 (0.33–1.25) | NS |
| Student | 168 (31.82) | 360 (68.18) | | 0.22 (0.16–0.3) | < .001 | 0.09 (0.05–0.17) | < .001 |
| Other¥ | 68 (43.59) | 88 (56.41) | | 0.37 (0.25–0.54) | < .001 | 0.21 (0.11–0.42) | < .001 |
| Religion | | | | | | | |
| Islam | 468 (45.53) | 560 (54.47) | | Ref | | Ref | |
| Hinduism | 86 (64.18) | 48 (35.82) | < .001 | 2.14 (1.48–3.12) | < .001 | 2.54 (1.36–4.74) | < .01 |
| Buddhists/Cristian | 10 (55.56) | 8 (44.44) | | 1.5 (0.59–3.82) | NS | 2.37 (0.4–14.02) | NS |
| Region | | | | | | | |
| Urban | 284 (55.47) | 228 (44.53) | < .001 | Ref | | Ref | |
| Rural | 280 (41.92) | 388 (58.08) | | 0.58 (0.46–0.73) | < .001 | 0.31 (0.2–0.48) | < .001 |

NS = not significant at 5% level; UOR = Unadjusted Odds Ratio; AOR = Adjusted Odds Ratio; Other¥ includes Housewife, Retired, and Unemployed; Other£ includes Divorced, Widowed, and Separated

respondents were 4 times (95% CI: 2.51–6.42) higher than the odds of having adequate knowledge among the married respondents. Among respondents who completed undergraduate or running, 62.75% had adequate knowledge, and among respondents with master's or higher degrees, 58.18% had adequate knowledge. However, only 15.66% of the respondents with less or equal HSC degrees (< = 12 grade) had adequate knowledge. The respondents with education level undergraduate and graduate had respectively 5 times (95% CI: 2.93–8.74) and 3 times (95% CI: 2.2–6.27) higher chance of having adequate knowledge than respondents with

an education level less or equal to HSC (grade 12). Respondents having higher incomes were more likely to have adequate knowledge about TB. People living in urban areas were more likely to have adequate knowledge. The respondents who live in rural areas had a 69% (95% CI: 0.2–0.48) lower chance of having adequate TB knowledge.

## Practices toward tuberculosis and associated factors

Nearly half of the participants preferred to visit health care centers (48.14%) as soon as they realized that their symptoms might be related to TB and wanted to take modern drugs (71%) (Table 2). Less than one-third of 1,180 respondents (31.19%) showed overall good practices (Table 4). According to bivariate analysis, 37.57% of males and 22.13% of females were doing good practices. According to findings from regression analysis, female respondents had a 41% (AOR = 0.59, 95% CI: 0.44–0.79) lower chance of doing good practices towards TB than males. The respondents aged 30–44 had a 1.41 (AOR = 1.41, 95% CI: 1.04–1.92) times higher chance of doing good practices than younger respondents (aged 18–29). Good practices were also observed among 28.26% of married and 41.46% of unmarried respondents. Unmarried respondents had a 1.46, (95% CI: 1.07–1.96) times higher likelihood of having good practices than married respondents. Among respondents with education level undergraduate and graduate, 30% had good practices while 21.69% of respondents with less or equal HSC degrees (< = 12 grade) had good practices. The odds of having good practices among the respondents with a graduate-level education were two times higher than respondents with less or equal HSC degrees. Among the respondents who believe in Hinduism, 55.22% had good practices toward TB, and for those who believe in Islam, 28.21% had good practices. The odds of having good practices towards TB among students and entrepreneurs/businesses, respectively, were 1.45 (95% CI: 0.99–2.12) and 1.93 (95% CI: 1.25–3.06) times higher than the odds of having good practices among service holders (govt/private).

## Attitude toward tuberculosis and associated factors

According to 79.32% (936) of the respondents, TB is a severe disease (Table 2). Almost half of the participants were afraid to get infected with TB. A large proportion (83%) of the respondents wanted to keep it secret when any family member gets TB. A significant portion of the respondents (54.92%) were unwilling to work with someone previously treated for TB. Also, a considerable proportion (44.75%) of the respondents had stigmatizing thoughts toward TB patients. Table 5 depicts that almost 45% of the respondents expressed a favorable attitude toward TB. The general population's attitude in Bangladesh toward TB and associated factors are also shown in Table 5. Gender was one of the significant factors of favorable attitudes toward TB. The odds of having a favorable attitude toward TB among females were 95% (95% CI: 0.03–0.07) lower than the odds of having a favorable attitude toward TB among males. The age of the respondents was also a significantly associated factor for having a favorable attitude toward TB. Middle-aged people were more likely to have a favorable attitude toward TB. More explicitly, respondents aged 30–40 had a 3.78 times (95% CI: 2.44–5.86) higher likelihood of having a favorable attitude toward TB than respondents aged 18–29. However, older (45–59) respondents had around 69% (95% CI: 0.16–0.6) lower chance of having a favorable attitude toward TB than the young respondents (aged 18–29). Marital status, education level, and income were highly correlated with the overall attitude of respondents toward TB. The odds of having a favorable attitude toward TB among the unmarried respondents were 3 times (95% CI: 1.93–4.73) higher than the odds of having a favorable attitude toward TB among the married respondents. The respondents with undergraduate and graduate-level education had respectively 6 times (95% CI: 3.77–11.38) and 1.96 times (95% CI: 1.16–3.29) higher chance of

**Table 4. Associated factors with practice toward TB among the participants.**

| Factors | Bivariate analysis | | | | | Multivariable analysis | |
|---|---|---|---|---|---|---|---|
| | Good Practice n (%) | Bad practice n (%) | *P*-value | UOR (95% CI) | *P*-value | AOR (95% CI) | *P*-value |
| Total | 368 (31.19) | 812 (68.81) | | | | | |
| Gender | | | | | | | |
| Male | 260 (37.57) | 432 (62.43) | < .001 | Ref | | Ref | |
| Female | 108 (22.13) | 380 (77.87) | | 0.47(0.36–0.61) | < .001 | 0.59 (0.44–0.79) | < .001 |
| Age | | | | | | | |
| 18–29 | 112 (27.72) | 292 (72.28) | | Ref | | Ref | |
| 30–44 | 200 (38.46) | 320 (61.54) | < .001 | 1.63(1.23–2.16) | < .01 | 1.41 (1.04–1.92) | < .05 |
| 45–59 | 52 (21.67) | 188 (78.33) | | 0.72(0.49–1.05) | NS | 0.88 (0.58–1.33) | NS |
| 75+ | 4 (25.00) | 12 (75.00) | | 0.87(0.27–2.75) | NS | 1.34 (0.34–5.26) | NS |
| Marital Status | | | | | | | |
| Married | 208 (28.26) | 528 (71.74) | | Ref | | Ref | |
| Unmarried | 136 (41.46) | 192 (58.54) | < .001 | 1.8(1.37–2.36) | < .001 | 1.45 (1.07–1.96) | < .05 |
| Other£ | 24 (20.69) | 92 (79.31) | | 0.66(0.41–1.07) | NS | 0.75 (0.43–1.3) | NS |
| Education | | | | | | | |
| Less or equal HSC (< = 12th grade) | 72 (21.69) | 260 (78.31) | | Ref | | | |
| Undergraduate | 124 (30.39) | 284 (69.61) | < .001 | 1.58(1.13–2.21) | < .05 | 0.97 (0.66–1.42) | NS |
| Master's or higher | 172 (30.09) | 268 (60.91) | | 2.32(1.68–3.2) | < .001 | 2.09 (1.42–3.08) | < .001 |
| Income | | | | | | | |
| Less than 30,000 | 92 (33.33) | 184 (66.67) | | Ref | | Ref | |
| 30,000–45,000 | 24 (13.64) | 152 (86.36) | | 0.32(0.19–0.52) | < .001 | 0.31 (0.18–0.53) | < .001 |
| 46,000–60,000 | 44 (27.50) | 116 (72.50) | < .001 | 0.76(0.49–1.16) | NS | 0.79 (0.49–1.26) | NS |
| 61,000–75,000 | 100 (35.71) | 180 (64.29) | | 1.11(0.78–1.58) | NS | 1.06 (0.72–1.56) | NS |
| 76,000 and above | 108 (37.50) | 180 (62.50) | | 1.2(0.85–1.7) | NS | 1 (0.67–1.51) | NS |
| Occupation | | | | | | | |
| Service holder (govt/private) | 104 (32.10) | 220 (67.90) | | Ref | | Ref | |
| Entrepreneur/business | 68 (39.53) | 104 (60.47) | <0.05 | 1.38(0.94–2.03) | 0.098 | 1.96 (1.25–3.06) | < .05 |
| Student | 160 (30.30) | 368 (69.70) | | 0.92(0.68–1.24) | NS | 1.45 (0.99–2.12) | < .05 |
| Other¥ | 36 (23.08) | 120 (76.92) | | 0.63(0.41–0.98) | < .05 | 0.72 (0.45–1.15) | NS |
| Religion | | | | | | | |
| Islam | 290 (28.21) | 738 (71.79) | | Ref | | Ref | |
| Hinduism | 74 (55.22) | 60 (44.78) | < .001 | 3.14(2.18–4.53) | < .001 | 2.57 (1.74–3.79) | < .001 |
| Buddhists/Cristian | 4 (22.22) | 14 (77.78) | | 0.73(0.24–2.23) | NS | 0.72 (0.23–2.27) | NS |
| Region | | | | | | | |
| Urban | 164 (32.03) | 348 (67.97) | NS | Not retained | | Not retained | |
| Rural | 204 (30.54) | 464 (69.46) | | | | | |

NS = not significant at 5% level; UOR = Unadjusted Odds Ratio; AOR = Adjusted Odds Ratio; Other¥ includes Housewife, Retired, Unemployed; Other£ includes Divorced, Widowed, and Separated

having a favorable attitude toward TB than respondents with an education level less or equal HSC (< = 12th grade). The respondents who live in rural areas had a 54% (95% CI: 0.3–0.69) lower chance of having a favorable attitude toward TB.

## Discussion

A large proportion (43%) of people in Bangladesh have internet access which is substantially increasing over time (19% increased between 2020 and 2021) [20, 28]. This study aimed to

**Table 5. Associated factors with attitude toward TB among the participants.**

| Factors | Bivariate analysis | | | | | Multivariable analysis | |
|---|---|---|---|---|---|---|---|
| | Favorable attitude n (%) | Unfavorable attitude n (%) | *P*-value | UOR (95% CI) | *P*-value | AOR (95% CI) | *P*-value |
| Total | 528 (44.75) | 652 (55.25) | | | | | |
| Gender | | | | | | | |
| Male | 464 (67.05) | 228 (32.95) | < .001 | Ref | | Ref | |
| Female | 64 (13.11) | 424 (86.89) | | 0.07 (0.05–0.1) | < .001 | 0.05 (0.03–0.07) | < .001 |
| Age | | | | | | | |
| 18–29 | 152 (37.62) | 252 (62.38) | | Ref | | Ref | |
| 30–44 | 332 (63.85) | 188 (36.15) | < .001 | 2.93 (2.24–3.83) | < .001 | 3.78 (2.44–5.86) | < .001 |
| 45–59 | 44 (18.33) | 196 (81.67) | | 0.37 (0.25–0.55) | < .001 | 0.31 (0.16–0.6) | < .001 |
| 75+ | 0 (0.00) | 16 (100.00) | | - | | - | |
| Marital Status | | | | | | | |
| Married | 280 (38.04) | 456 (61.96) | | Ref | | Ref | |
| Unmarried | 232 (70.73) | 96 (29.27) | < .001 | 3.94 (2.97–5.21) | < .001 | 3.02 (1.93–4.73) | < .001 |
| Other[£] | 16 (13.79) | 100 (86.21) | | 0.26 (0.15–0.45) | < .001 | 0.55 (0.25–1.23) | NS |
| Education | | | | | | | |
| Less or equal HSC (< = 12[th] grade) | 52 (15.66) | 280 (84.34) | | Ref | | Ref | |
| Undergraduate | 256 (62.75) | 152 (37.25) | < .001 | 9.07 (6.34–12.97) | < .001 | 6.55 (3.77–11.38) | < .001 |
| Master's or higher (Graduate) | 220 (50.00) | 220 (50.00) | | 5.38 (3.79–7.64) | < .001 | 1.95 (1.16–3.29) | < .01 |
| Income | | | | | | | |
| Less than 30,000 | 96 (34.78) | 180 (65.22) | | Ref | | Ref | |
| 30,000–45,000 | 28 (15.91) | 148 (84.09) | | 0.35 (0.22–0.57) | < .001 | 0.38 (0.2–0.74) | < .01 |
| 46,000–60,000 | 56 (35.00) | 104 (65.00) | < .001 | 1.01 (0.67–1.52) | NS | 1.22 (0.64–2.33) | NS |
| 61,000–75,000 | 120 (42.86) | 160 (57.14) | | 1.41 (1–1.98) | < .05 | 1.25 (0.72–2.18) | NS |
| 76,000 and above | 228 (79.17) | 60 (20.83) | | 7.13 (4.89–10.39) | < .001 | 9.67 (4.86–19.27) | < .001 |
| Occupation | | | | | | | |
| Service holder (govt/private) | 200 (61.73) | 124 (38.27) | | Ref | | Ref | |
| Entrepreneur/business | 96 (55.81) | 76 (44.19) | | 0.78 (0.54–1.14) | NS | 0.68 (0.36–1.28) | NS |
| Student | 176 (33.33) | 352 (66.67) | < .001 | 0.31 (0.23–0.41) | < .001 | 0.19 (0.11–0.32) | < .001 |
| Other[¥] | 56 (35.90) | 100 (64.10) | | 0.35 (0.23–0.52) | < .001 | 0.17 (0.09–0.32) | < .001 |
| Religion | | | | | | | |
| Islam | 442 (43.00) | 586 (57.00) | | Ref | | Ref | |
| Hinduism | 76 (56.72) | 58 (43.28) | < .01 | 1.74 (1.21–2.5) | < .05 | 1.65 (0.91–2.99) | NS |
| Buddhists/Cristian | 10 (55.56) | 8 (44.44) | | 1.66 (0.65–4.23) | NS | 3.14 (0.46–21.69) | NS |
| Region | | | | | | | |
| Urban | 264 (51.56) | 248 (48.44) | < .001 | Ref | | Ref | |
| Rural | 264 (39.52) | 404 (60.48) | | 0.61 (0.49–0.77) | < .001 | 0.46 (0.3–0.69) | < .001 |

NS = not significant at 5% level; UOR = Unadjusted Odds Ratio; AOR = Adjusted Odds Ratio; Other¥ includes Housewife, Retired, Unemployed; Other£ includes Divorced, Widowed, and Separated

inspect the level of knowledge, good attitudes, and practices among the general people who have internet access by circulating a survey link through social media and other electronic platforms. A total of 1,180 online users completed the survey and 58.64% of the participants were male. About 80% of the study participants fell in the age category of 18–44 years, which indicates a young study population. However, this age group is of immense interest because, according to Bangladesh's national tuberculosis program, three-quarters of TB cases in Bangladesh were in the age bracket of 18–45 years [13]. Only 28.14% of study participants had HSC

or less ($<= 12^{th}$ grade) education level, and a substantial portion of the participants were current students (44.75%). The above socio-demographic characteristics are understandable due to the nature of the survey. Participants also showed a good level of urban-rural balance (rural-56%). Young, educated, and student populations understandably have greater access to social media and electronic platforms.

Almost all survey participants in this study had heard of tuberculosis, which is consistent with research undertaken in Nigeria, India, Pakistan, and Lesotho [1, 29–31]. The primary sources of information about TB were TV/Radio/Newspapers, Leaflets/Posters/Signboards/Billboards, health professionals, the internet, teachers/religious leaders, and family/friends/relatives. The sources of information found in this study are parallel with a previous study conducted among adult TB patients in Dhaka Bangladesh [13]. However, a study was conducted to inspect Bangladeshi mothers' knowledge of childhood tuberculosis and found a large proportion of mothers (84%) had no idea about childhood TB [32]. In this study, we found that just 47.8% of respondents (social media users) possessed sufficient overall knowledge of tuberculosis. A sizable number (71%) of participants knew that tuberculosis germs or bacteria are primarily responsible for TB infection. However, this knowledge level is lower than similar studies conducted in Malawi, Ethiopia, and India (90%, 81.7%, and 81%, respectively) and higher than the study taken part among non-medical university students in Bangladesh (42%) [33]. In addition, less than half of the respondents (42%) were aware of the correct mode of transmission of tuberculosis (TB can be transmitted from person to person via coughing or sneezing). A lower level of knowledge regarding TB transmission among women also was estimated in a previous study (7%) [34]. These findings indicate a substantially low level of knowledge about TB among social media users in Bangladesh compared to the findings of other studies conducted among general people.

The participants' knowledge regarding signs of TB was insufficient. Less than half of the respondents knew the common symptoms of TB infection (cough for more than three weeks, persistent fever, nighttime sweating, and weight loss). This finding is consistent with similar studies conducted in Bangladesh and Lesotho [1, 35]. On the other hand, respondents were well informed about the availability of tuberculosis treatment, with more than three-fourths of respondents being aware of the curability of the disease and the availability of free treatment in Bangladesh. This is consistent with other global studies in Brazil, India, and Tanzania [36–38]. Female social media users had a significantly lower likelihood of possessing adequate knowledge of TB, and middle-aged (30–40 years) people had a significantly higher likelihood of having adequate knowledge than younger and older people. These two results are coherent with a nationwide study conducted in Bangladesh [39]. Social media users who had higher incomes were more likely to have adequate knowledge about tuberculosis, while social media users who live in rural areas had a considerably lower chance of having an adequate understanding of tuberculosis.

Less than a third (31.19%) of 1,180 study participants demonstrated overall good practices (Table 4). This is an abysmal level of good practices even compared to the estimate of slum dwellers in Nigeria (48.8%) [40] and some other studies in Gambia, and Pakistan [41, 42]. Gender, age, marital status, education, and religion were significantly associated with good practices toward TB with females, married, younger and older, less educated, and Muslim people showing poor practices toward TB. Although poor practices were found among social media users more prevalent in this study than in many others, the risk factors for poor practices discovered in this study are consistent with other studies among different types of populations [41–43].

Fewer than half of the participants expressed a favorable attitude about tuberculosis (44.75%) (Table 5). A lack of proper knowledge about TB might be why this study found such

a low level of positive attitudes among social media users toward TB. The estimate of favorable attitude is inconsistent with other identical studies among different populations showing a higher positive attitude toward TB than this study [36, 40–43]. A considerable proportion of respondents wanted to keep it secret if any family member gets TB, were unwilling to work with one previously treated for TB, and had stigmatizing thoughts about TB patients. Like knowledge and practices, favorable attitudes among social media users had similar risk factors. Gender, age, education level, marital status, and region (urban/rural) were significantly associated with favorable attitudes toward TB. Unmarried and undergraduate/graduate level respondents displayed 3 times and 2–6 times higher likelihood of having favorable attitudes toward TB, respectively. Female social media users and rural social media users showed a 95% and 54% lower likelihood of possessing positive attitudes, respectively. Covariates identified for attitudes towards TB are accordant with other studies conducted in developing countries like Bangladesh [1, 13, 44, 45]. However, Luba et al. and some other studies found higher positive attitudes among females and married people, which is inconsistent with this study [1, 13].

Social media websites and platforms can boost professional growth and advancement as well as individual and public health when used properly and sensibly [46]. Since a large number of people in Bangladesh, about 36 million, are using social media platforms such as Facebook, YouTube, Instagram, WhatsApp, IMO, etc., these platforms can be used as a medium to elevate people's tuberculosis-related knowledge, attitudes, and behaviors. TB prevalence can be lowered in Bangladesh by creating awareness among general people using social media. This is a very attractive and effective platform to convey information regarding TB disease, its signs and symptoms, how to prevent it, and how to detect and treat it along with other public health issues [47]. There is evidence that TB is significantly prevalent among the young population (18–45 years old) in Bangladesh who are mainly active users of social media [13]. If this targeted population can be trained for TB, we can expect a substantial improvement in TB cases and mortality in Bangladesh. It is recommended that Bangladesh Govt., policymakers, and public health experts create and share scientific content on TB and other diseases on social media to educate people.

One of the strengths of our study is that to our knowledge, for the first time in Bangladesh, we conducted a study to assess the knowledge, attitudes, and practices towards TB and their associated factors among a large proportion of general people, who have internet access. Also, our sample size was significantly greater than some other relevant studies. However, this study has some limitations. The low response rate (30%), which is common for voluntary online surveys, may result in sampling bias. Moreover, since the responses were self-reported, information bias may also occur. And since the study participants were only social media/internet users, the findings cannot be generalized to the general population of Bangladesh. However, the findings give an idea of the knowledge, attitudes, and practices among the general population in Bangladesh. Fig 2 shows that mainly men (around 53% of the men) and people who have higher education (70% among those who have completed higher study, 49% of those who have completed secondary education whereas only 6% and 20% of those who have no education and completed primary) dominantly have internet access in Bangladesh [20] which implies that the estimated knowledge, practices, and attitudes may be poorer among the general population compared to this study's findings.

## Conclusion

This study revealed poor knowledge, attitudes, and practices toward TB among social media users in Bangladesh. Less than half of the study participants showed sufficient knowledge, good practices, and favorable attitudes toward TB. These findings are poorer than most other

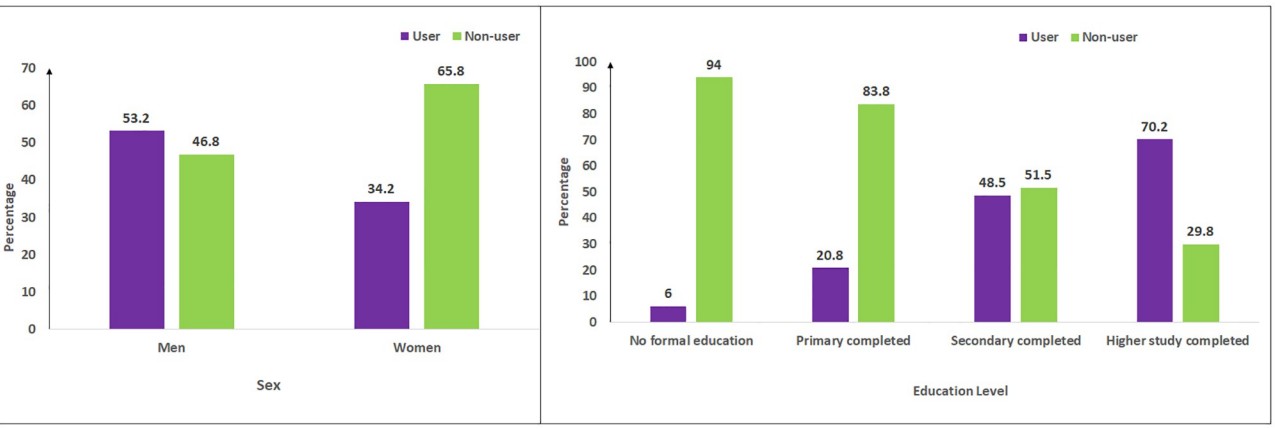

**Fig 2. The distribution of internet users and non-users in Bangladesh across sex and education levels (Source: Bangladesh National ICT Household Survey [20]).**

study findings conducted in developing country settings. Females, older, illiterate/less educated, married, and rural people were more vulnerable to having poor knowledge, attitudes, and practices toward TB. Bangladesh government and policymakers should design internet-based programs and interventions to improve knowledge, attitudes, and practices about TB among social media users in a bid to achieve the End TB goals, including a 95% reduction in TB mortality and a 90% reduction in TB incidence by 2035 compared to 2015 levels. In this study, participants had internet/social media access and are supposed to have better knowledge, attitudes, and practices toward TB. Hence, the findings should be interpreted and generalized with caution. Future studies should be more representative of the general population of Bangladesh.

## Supporting information

**S1 Questionnaire.**
(DOCX)

**S1 Checklist.**
(DOCX)

## Acknowledgments

We are grateful to all who spent their valuable time participating in the survey voluntarily and sharing the link with others. We are also grateful to those who, despite not being eligible, shared the link and inspired others to participate.

## Author Contributions

**Conceptualization:** Sultan Mahmud, Md Mohsin.

**Data curation:** Sultan Mahmud, Md Mohsin, Saddam Hossain Irfan, Ariful Islam.

**Formal analysis:** Sultan Mahmud, Md Mohsin, Saddam Hossain Irfan.

**Investigation:** Sultan Mahmud.

**Methodology:** Sultan Mahmud, Abdul Muyeed, Ariful Islam.

**Resources:** Md Mohsin.

**Software:** Sultan Mahmud, Md Mohsin, Saddam Hossain Irfan.

**Supervision:** Sultan Mahmud, Md Mohsin.

**Validation:** Sultan Mahmud, Md Mohsin, Ariful Islam.

**Visualization:** Sultan Mahmud.

**Writing – original draft:** Sultan Mahmud, Md Mohsin, Abdul Muyeed, Ariful Islam.

**Writing – review & editing:** Sultan Mahmud, Md Mohsin, Saddam Hossain Irfan, Abdul Muyeed, Ariful Islam.

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
