## [Decision Letter · Decision Letter 0]

29 Jun 2022

PONE-D-22-04571

Knowledge, Attitude, Practices, and Determinants of Them towards Tuberculosis in Bangladesh: A Cross-sectional Study

PLOS ONE

Dear Dr. Mohsin,

Thank you for submitting your manuscript to PLOS ONE. After careful consideration, we feel that it has merit but does not fully meet PLOS ONE’s publication criteria as it currently stands. Therefore, we invite you to submit a revised version of the manuscript that addresses the points raised during the review process.

We look forward to receiving your revised manuscript.

Kind regards,

Leeberk Raja Inbaraj, MD

Academic Editor

PLOS ONE

Reviewers' comments:

Reviewer's Responses to Questions

**Comments to the Author**

1. Is the manuscript technically sound, and do the data support the conclusions?

Reviewer #1: Partly

2. Has the statistical analysis been performed appropriately and rigorously? 

Reviewer #1: Yes

3. Have the authors made all data underlying the findings in their manuscript fully available?

Reviewer #1: Yes

4. Is the manuscript presented in an intelligible fashion and written in standard English?

Reviewer #1: Yes

5. Review Comments to the Author

Reviewer #1: This manuscript describes an online survey on the Knowledge, Attitude and Practices related to Tb in Bangladesh. The topic is important since Knowledge and attitude play a key role in the process of disease control. The authors have described the study methodology well and the topic is of utmost importance. I have a few queries and concerns regarding the study as mentioned below:

Major concerns

1. Representativeness of the study – Online surveys may not be representative of the general population. This is especially true in some areas where internet connectivity and social media usage may be very low.

Representativeness is best achieved through probability sampling and two key components are required for probability sampling. I) Known non-zero chance of selecting any member of the target population. II)Random selection of members

Both these factors are not present in this study. Hence the results cannot be projected to the general population. Several methods have been evolved to address representativeness in online surveys such as probability or open online panels, weighting, quota or river sampling, sample matching etc. If any such methods were used in this study please mention that.

Some statistics on the usage of social media, the proportion of the population with access to the internet and social media etc may help judge the representativeness to an extent. A comparison of the sociodemographic factor distribution of this study sample with that of the general population may reveal insights regarding the representativeness of the study population.

2. Bias- In Section 4.1: Strengths and Limitations of the study- only information bias due to self-reporting is addressed.

The study also has several factors for selection/sampling bias, specifically volunteer bias and low response rates. Social media users with some knowledge about Tb may have been more likely to click the link for the survey. If any methods were used to improve this as mentioned in Question 1 please mention that. The lower the response rate the higher the risk of sampling bias.

3. Development and Validation of the questionnaire – Is this an existing questionnaire or an adapted version of an existing questionnaire. Otherwise, the process of development and validation of the questionnaire has to be mentioned in more detail. The methods section only mentions that the “questionnaire was validated by several experts and pilot surveys”. How the questionnaire was developed and validated is important if it is a new questionnaire.

4. The Discussion needs to include an assessment on the representativeness of this study for the general population of Bangladesh as mentioned before. Additionally, while there are very few KAP studies on the topic among the general population in Bangladesh, several studies are available on specific subgroups such as students, industrial workers etc. Comparison with these studies may give further insights.

Minor

5. The sample size of the study is much higher than the calculated sample size. If sample size calculation was done retrospectively, then power calculation may be a better option.

6. The introduction can be written with more clarity.

Eg On Page 2, paragraph 2 it is mentioned that “There are also a lot of misconceptions concerning the aetiology and mode of transmission in Bangladesh [13]. TB is thought to be inherited in some locations [14–16].” The juxtaposition of the two sentences suggests that the misconception that Tb is inherited exists in Bangladesh. However, all the studies referenced are from the African continent.

Paragraphs 1 and 2 on Page 1 (introduction) are slightly confusing and repetitive with Tb statistics from different years mentioned in different places. The first paragraph mentions statistics from Global Tb Report 2014 and 2015, while the second paragraph uses the Global Tb report 2021 statistics. If the intention is to show the change in Tb incidence or mortality that is not conveyed here.

Eg. Paragraph 1 mentions “ Globally, 10.4 million people were reported to have contracted tuberculosis in 2015, with 1.8 million people dying from the disease”

While in paragraph 2 it is mentioned “Globally, 10 million individuals were infected with tuberculosis in 2019, with 79% of those infected living in the 30 high-burden countries and 1.2 million people dying from the disease”.

7. In Section 2.3, subsections 2.3.2 and 2.3.3 mention all the questions that were part of the questionnaire. This is redundant considering that Table 3 has all the questions and answers along with the proportion of responses. A brief description of the questionnaire along with any methods used to reduce duplicates etc should be sufficient.

8. In section 2.5 the second paragraph describes the statistical analysis. While the term 'multivariate logistic regression' is used interchangeably with 'multivariate analysis' in some publications, the term “multivariate logistic regression” is usually used for models with multiple outcome/dependent variables. In this manuscript, the term multiple or multivariable logistic regression may be more appropriate. (Please see Ref: Hidalgo B, Goodman M. Multivariate or Multivariable Regression? Am J Public Health. 2013 Jan 1;103(1):39–40.)

9. Section 2.6 is repetitive. All the points have already been mentioned in sections 2.1 and 2.4.

10. Some minor grammatical errors need correction. Please use editing software. Eg. The term “general people” should be replaced with “the general population”.

6. PLOS authors have the option to publish the peer review history of their article (what does this mean?). If published, this will include your full peer review and any attached files.

Reviewer #1: **Yes: **Martina Shalini Arul Joseph

---

## [Author Response · Author response to Decision Letter 0]

6 Jul 2022

Response to Editor/ Reviewer: 

Reviewer point #1: “Representativeness of the study – Online surveys may not be representative of the general population. This is especially true in some areas where internet connectivity and social media usage may be very low.

Representativeness is best achieved through probability sampling and two key components are required for probability sampling. I) Known non-zero chance of selecting any member of the target population. II)Random selection of members

Both these factors are not present in this study. Hence the results cannot be projected to the general population. Several methods have been evolved to address representativeness in online surveys such as probability or open online panels, weighting, quota or river sampling, sample matching etc. If any such methods were used in this study please mention that.

Some statistics on the usage of social media, the proportion of the population with access to the internet and social media etc may help judge the representativeness to an extent. A comparison of the sociodemographic factor distribution of this study sample with that of the general population may reveal insights regarding the representativeness of the study population.”

Author response #1: Thank you so much for catching the issue. We completely agree with you that the results of our study can not be generalized to the general population of Bangladesh. However, we have interpreted our findings for the people who have internet access in the revised manuscript. Hence, we have replaced the word “general population/ general people” with people who have internet access. In addition, we have added a comparative distribution across gender and education levels (Figure 5) of internet users and non-users in Discussion Section to get an idea about knowledge, attitudes, and practices toward TB among the general population.

Reviewer point #2: “Bias- In Section 4.1: Strengths and Limitations of the study- only information bias due to self-reporting is addressed.

The study also has several factors for selection/sampling bias, specifically volunteer bias and low response rates. Social media users with some knowledge about Tb may have been more likely to click the link for the survey. If any methods were used to improve this as mentioned in Question 1 please mention that. The lower the response rate the higher the risk of sampling bias.”

Author response #2: Thank you for the legit suggestion. We have updated Strengths and Limitations accordingly.

Reviewer point #3: “Development and Validation of the questionnaire – Is this an existing questionnaire or an adapted version of an existing questionnaire. Otherwise, the process of development and validation of the questionnaire has to be mentioned in more detail. The methods section only mentions that the “questionnaire was validated by several experts and pilot surveys”. How the questionnaire was developed and validated is important if it is a new questionnaire.”

Author response #3: Thank you for the comment. Yes, we have used an adapted version of existing questionnaires. We have mentioned it in subsection 2.1 in the revised manuscript.

Reviewer point #4: “The Discussion needs to include an assessment on the representativeness of this study for the general population of Bangladesh as mentioned before. Additionally, while there are very few KAP studies on the topic among the general population in Bangladesh, several studies are available on specific subgroups such as students, industrial workers, etc. Comparison with these studies may give further insights.”

Author response #4: As we mentioned above, we are not generalizing our findings to the general population without any caution in the revised manuscript. we have also added some statistics regarding our study participants, people who have internet access. Moreover, we have added some comparative views for our findings with existing studies in the Discussion Section. 

Reviewer point #5: “The sample size of the study is much higher than the calculated sample size. If sample size calculation was done retrospectively, then power calculation may be a better option.”

Author response #5: Thank you for your comments. The main purpose of our study was to make inferences about the level of knowledge, attitudes, and practices based on the sample information. As we don’t intend to reject some null hypothesis (if it happens to be false), we are concerned about the precision of any estimate from the sample but not about the statistical power. However, we found that the absolute precision for the sample we collected is 5.2%. This has been added in 3.1 subsections. 

Reviewer point #6: “The introduction can be written with more clarity.

Eg On Page 2, paragraph 2 it is mentioned that “There are also a lot of misconceptions concerning the aetiology and mode of transmission in Bangladesh [13]. TB is thought to be inherited in some locations [14–16].” The juxtaposition of the two sentences suggests that the misconception that Tb is inherited exists in Bangladesh. However, all the studies referenced are from the African continent.

Paragraphs 1 and 2 on Page 1 (introduction) are slightly confusing and repetitive with Tb statistics from different years mentioned in different places. The first paragraph mentions statistics from Global Tb Report 2014 and 2015, while the second paragraph uses the Global Tb report 2021 statistics. If the intention is to show the change in Tb incidence or mortality that is not conveyed here.

Eg. Paragraph 1 mentions “ Globally, 10.4 million people were reported to have contracted tuberculosis in 2015, with 1.8 million people dying from the disease”

While in paragraph 2 it is mentioned “Globally, 10 million individuals were infected with tuberculosis in 2019, with 79% of those infected living in the 30 high-burden countries and 1.2 million people dying from the disease”.”

Author response #6: We agree with you that there were some ambiguities in the introduction section. Thank you very much for identifying them. We have removed the sentence “TB is thought to be inherited in some locations [14–16]”, because the references are based in African countries and we didn’t find such references based in Bangladesh. 

 You can see that we revised the write-up to reduce the redundancy of similar statistics. However, there are still few stats from the different years to show that unsatisfactory progress has been made in controlling TB in the last few years. 

Reviewer point #7: “In Section 2.3, subsections 2.3.2 and 2.3.3 mention all the questions that were part of the questionnaire. This is redundant considering that Table 3 has all the questions and answers along with the proportion of responses. A brief description of the questionnaire along with any methods used to reduce duplicates etc should be sufficient.”

Author response #7: Thank you for raising this point. As you suggested, we have removed subsections 2.3.2 and 2.3.3 from Section 2.3 and added subsection 2.5, which described in detail the data management process. 

Reviewer point #8: “In section 2.5 the second paragraph describes the statistical analysis. While the term 'multivariate logistic regression' is used interchangeably with 'multivariate analysis' in some publications, the term “multivariate logistic regression” is usually used for models with multiple outcome/dependent variables. In this manuscript, the term multiple or multivariable logistic regression may be more appropriate. (Please see Ref: Hidalgo B, Goodman M. Multivariate or Multivariable Regression? Am J Public Health. 2013 Jan 1;103(1):39–40.)”

Author response #8: I would like to thank you for catching this issue. This was a serious typo. We have corrected it in the revised manuscript. 

Reviewer point #9: “Section 2.6 is repetitive. All the points have already been mentioned in sections 2.1 and 2.4.”

Author response #9: Section 2.6 has been deleted.

Reviewer point #10: “Some minor grammatical errors need correction. Please use editing software. Eg. The term “general people” should be replaced with “the general population””

Author response #10: Thank you for your suggestion. The manuscript has been checked for grammatical issues with Grammarly Premium services. The corrections have been made. 

In addition, we went through the whole article for similar types of typos and editing mistakes. We believe that the changes have improved the revised manuscript, which you will find updated.

---

## [Decision Letter · Decision Letter 1]

11 Aug 2022

PONE-D-22-04571R1Knowledge, Attitude, Practices, and Determinants of Them towards Tuberculosis in Bangladesh: A Cross-sectional StudyPLOS ONE

Dear Dr. Mohsin,

Thank you for submitting your manuscript to PLOS ONE. After careful consideration, we feel that it has merit but does not fully meet PLOS ONE’s publication criteria as it currently stands. Therefore, we invite you to submit a revised version of the manuscript that addresses the points raised during the review process.

We look forward to receiving your revised manuscript.

Kind regards,

Leeberk Raja Inbaraj, MD

Academic Editor

PLOS ONE

Journal Requirements:

Reviewers' comments:

Reviewer's Responses to Questions

**Comments to the Author**

1. If the authors have adequately addressed your comments raised in a previous round of review and you feel that this manuscript is now acceptable for publication, you may indicate that here to bypass the “Comments to the Author” section, enter your conflict of interest statement in the “Confidential to Editor” section, and submit your "Accept" recommendation.

Reviewer #1: (No Response)

2. Is the manuscript technically sound, and do the data support the conclusions?

Reviewer #1: Yes

3. Has the statistical analysis been performed appropriately and rigorously? 

Reviewer #1: Yes

4. Have the authors made all data underlying the findings in their manuscript fully available?

Reviewer #1: Yes

5. Is the manuscript presented in an intelligible fashion and written in standard English?

Reviewer #1: No

6. Review Comments to the Author

Reviewer #1: This is a study describing the knowledge, attitude and practices regarding Tuberculosis among social media users in Bangladesh. The results are important since social media has become the primary source of all information and misinformation for a large proportion of the population. The suggestions made in the previous review have been mostly addressed. I have a few more concerns:

1. Section 2.1 line 6 mentions that participants "were requested to help those who do not have access to the internet or social media". What kind of help was suggested? If these "others" had no access to internet, that would imply that devices with internet access had to be shared. However, in section 2.5 it is mentioned that duplicate submissions were identified and dropped based on the device ids.

2. There are still some grammatical and typographical errors. Eg. device is spelled devise in section 2.5.

3. References: Some references are not in the correct format. Please check the correct format for reports and websites

Rather than attempt to extrapolate the results to the general population, the authors can focus on the interpretation of these results among social media users and their implications. The discussion can be rewritten with focus on the interpretation of this study's results and how they can inform future targeted health education using social media.

7. PLOS authors have the option to publish the peer review history of their article (what does this mean?). If published, this will include your full peer review and any attached files.

Reviewer #1: **Yes: **Martina Shalini Arul Joseph

---

## [Author Response · Author response to Decision Letter 1]

16 Aug 2022

Responses to reviewers:

Reviewer point#1: Section 2.1 line 6 mentions that participants "were requested to help those who do not have access to the internet or social media". What kind of help was suggested? If these "others" had no access to internet, that would imply that devices with internet access had to be shared. However, in section 2.5 it is mentioned that duplicate submissions were identified and dropped based on the device ids.

Author response#1: Thank you for identifying this significant discrepancy. Actually, our initial plan was to request participants to help others who don’t have internet/smart device access. However, later, we decided to survey only those who have internet/social media access. The sentence was taken from our initial study plan doc, and regrettably, we kept this sentence by mistake. The statement in section 2.5 about removing duplicate submissions based on device id is correct.

Reviewer point#2: There are still some grammatical and typographical errors. Eg. device is spelled devise in section 2.5.

Author response#2: We are sorry for the grammatical and typographical errors. We have gone through the entire manuscript with extensive care and corrected all the errors we identified. 

Reviewer point#3: References: Some references are not in the correct format. Please check the correct format for reports and websites

Author response#3: Thank you so much for identifying the discrepancies in the references. We have corrected the discrepancies in the revised manuscript.

Reviewer additional point: Rather than attempt to extrapolate the results to the general population, the authors can focus on the interpretation of these results among social media users and their implications. The discussion can be rewritten with focus on the interpretation of this study's results and how they can inform future targeted health education using social media.

Author response: Thank you for your insightful comment, we really appreciate it. We have rewritten the discussion section following your instructions.

---

## [Decision Letter · Decision Letter 2]

30 Aug 2022

PONE-D-22-04571R2Knowledge, Attitude, Practices, and Determinants of them toward Tuberculosis in Bangladesh: A Cross-sectional StudyPLOS ONE

Dear Dr. Mohsin,

Thank you for revising  your manuscript . The manuscript has improved significantly, however, the reviewer has given a few suggestions to further strengthen the quality of the paper .Therefore, we invite you to submit a revised version of the manuscript that addresses the points raised during the review process. The manuscript is likely to be accepted after the revision. 

We look forward to receiving your revised manuscript.

Kind regards,

Leeberk Raja Inbaraj, MD

Academic Editor

PLOS ONE

Journal Requirements:

Reviewers' comments:

Reviewer's Responses to Questions

**Comments to the Author**

1. If the authors have adequately addressed your comments raised in a previous round of review and you feel that this manuscript is now acceptable for publication, you may indicate that here to bypass the “Comments to the Author” section, enter your conflict of interest statement in the “Confidential to Editor” section, and submit your "Accept" recommendation.

Reviewer #1: (No Response)

2. Is the manuscript technically sound, and do the data support the conclusions?

Reviewer #1: Yes

3. Has the statistical analysis been performed appropriately and rigorously? 

Reviewer #1: Yes

4. Have the authors made all data underlying the findings in their manuscript fully available?

Reviewer #1: Yes

5. Is the manuscript presented in an intelligible fashion and written in standard English?

Reviewer #1: Yes

6. Review Comments to the Author

Reviewer #1: The authors have made all suggested changes. I have a few suggestions.

The title may be better if including the term internet users or social media users.

In socio-demographic characteristics(Section3.1)- is there a need to include calculated precision?

In Table 1 - no need to include Consent as a variable.

The article can be further shortened, figures 2,3,4 are not necessary. Similarly the data in table 2 can be included in tables 4,5,6 instead of a separate table.

In the discussion section comparing the demographic characteristics of the general population with that of this study may be useful. The Discussion can be streamlined further . In the Discussion Paragraph 5 on Pg14 states TB is significantly prevalent among the young population in Bangladesh who are mainly active users of social media. This statement is abrupt and requires either reference or further explanation. Or this statement can be included in the first paragraph where the importance of the 18-45 years age group is mentioned.

Please check format for references for reports and databases. Reports as references year of publication has to be mentioned (eg. Ref no. 24). For databases references should be according to ICJME guidelines(e.g. Ref: 22).

Please see samples in https://www.nlm.nih.gov/bsd/uniform_requirements.html

7. PLOS authors have the option to publish the peer review history of their article (what does this mean?). If published, this will include your full peer review and any attached files.

Reviewer #1: No

---

## [Author Response · Author response to Decision Letter 2]

5 Sep 2022

Responses to reviewers:

Reviewer point#1: The authors have made all suggested changes. I have a few suggestions.

The title may be better if including the term internet users or social media users.

Author response#1: Thank you for your suggestion. The title now includes social media users to specify the study participants.

Reviewer point#2: In socio-demographic characteristics(Section3.1)- is there a need to include calculated precision?

Author response#2: Yes, you are absolutely right. We have removed it. 

Reviewer point#3: In Table 1 - no need to include Consent as a variable.

Author response#3: This variable from Table 1 has been removed.

Reviewer point #4: The article can be further shortened, figures 2,3,4 are not necessary. Similarly, the data in table 2 can be included in tables 4,5,6 instead of a separate table.

Author response #4: The article has been shortened significantly following the above suggestions. Thank you for the suggestions. 

Reviewer point#5: In the discussion section comparing the demographic characteristics of the general population with that of this study may be useful. The Discussion can be streamlined further. In the Discussion Paragraph 5 on Pg14 states TB is significantly prevalent among the young population in Bangladesh who are mainly active users of social media. This statement is abrupt and requires either reference or further explanation. Or this statement can be included in the first paragraph where the importance of the 18-45 years age group is mentioned.

Author response #5: We agree with you that the sentence was abrupt without any references. We have revised the wording to make it coherent and also added a reference. We think the discussion section is good overall. 

Reviewer point#6: Please check format for references for reports and databases. Reports as references year of publication has to be mentioned (eg. Ref no. 24). For databases references should be according to ICJME guidelines(e.g. Ref: 22).

Author response #6: Thank you for identifying the issue and also for sharing the guidelines. We have modified them following the guidelines.

---

## [Editor Report · Decision Letter 3]

14 Sep 2022

Knowledge, Attitude, Practices, and Determinants of them toward Tuberculosis among Social Media Users in Bangladesh: A Cross-sectional Study

PONE-D-22-04571R3

Dear Dr. Mohsin,

We’re pleased to inform you that your manuscript has been judged scientifically suitable for publication and will be formally accepted for publication once it meets all outstanding technical requirements.

Kind regards,

Leeberk Raja Inbaraj, MD

Academic Editor

PLOS ONE
---

## [Editor Report · Acceptance letter]

2 Oct 2022

PONE-D-22-04571R3 

Knowledge, Attitude, Practices, and Determinants of them toward Tuberculosis among Social Media Users in Bangladesh: A Cross-sectional Study 

Dear Dr. Mohsin:

I'm pleased to inform you that your manuscript has been deemed suitable for publication in PLOS ONE. Congratulations! Your manuscript is now with our production department. 

Kind regards, 

on behalf of

Dr. Leeberk Raja Inbaraj 

Academic Editor

PLOS ONE